# DNA Barcoding of Cold-Water Coral-Associated Ophiuroid Fauna from the North Atlantic

**Angelina Eichsteller** [1,2,*], **James Taylor** [3], **Sabine Stöhr** [4], **Saskia Brix** [3,*] **and Pedro Martìnez Arbizu** [1,2]

1 Senckenberg am Meer, German Centre for Marine Biodiversity Research (DZMB), Südstrand 44, 26382 Wilhelmshaven, Germany; pedro.martinez@senckenberg.de

2 Marine Biodiversität, FKV-IBU, Universität Oldenburg, Carl-von-Ossietzky Strasse 15, 26129 Oldenburg, Germany

3 Senckenberg am Meer, German Centre for Marine Biodiversity Research (DZMB) c/o Biocenter Grindel, University of Hamburg, Martin-Luther-King-Platz 3, 20146 Hamburg, Germany; james.taylor@senckenberg.de

4 Department of Zoology, Swedish Museum of Natural History, Frescativ. 40, 10405 Stockholm, Sweden; sabine.stohr@nrm.se

* Correspondence: angelina.eichsteller@gmail.com (A.E.); sbrix@senckenberg.de (S.B.)

**Abstract:** In this study we focus on the ophiuroid species associated with cold-water corals south of Iceland. The specimens were sampled with the ROV Phoca (GEOMAR) in three different areas, during the recent expedition MSM75 connected to the IceAGE_RR (Icelandic marine Animals: Genetics and Ecology_Reykjanes Ridge hydrothermal vent activity) project. In each area, several corals were sampled and the ophiuroid specimens identified to the species level. The integrative taxonomic approach, based on morphological characters and DNA barcoding with COI of the collected ophiuroids, revealed five species that live on corals: *Ophiomitrella clavigera* (Ljungman, 1865); *Ophiomyxa serpentaria* (Lyman, 1883); *Ophiacantha cuspidata* (Lyman, 1879); *Ophiactis abyssicola* (M. Sars, 1861); and *Ophiolebes bacata* Koehler, 1921. Some of the sampled deep-sea corals exclusively host the species *O. clavigera*. The collected species are therefore associated with different corals but do not demonstrate a species-specific distribution. The video data support the integrative taxonomy and confirm the ecological evidence.

**Keywords:** echinoderms; brittle stars; Iceland; Reykjanes Ridge; COI; species delimitation; ROV; host preference

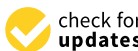



## 1. Introduction

The Nordic seas surrounding Iceland present geological and ecological interests and have been intensively studied during the last four decades [1–4]. The marine region around Iceland is indicated by a submarine mountain chain, the Greenland–Scotland-Ridge, and several water masses. The ridge forms a physical barrier that clearly separates the Arctic deep-sea basins from the North Atlantic ones, and its complex topography influences marine habitats. Cold, deep-water currents engulf Iceland from the western and eastern side with a north–south orientation. In contrast to this cold deep-water flow, warmer surface waters circulate around Iceland in a southwest to northeast direction [5–8]. The BIOFAR (1987–1992) [9] and BIOICE projects (1992–2004) [1] were the first extensive studies focusing on an inventorial effort of Icelandic benthic marine invertebrates. The follow-up project, IceAGE (Icelandic marine Animals: Genetics and Ecology, 2011–present), was then funded under a similar premise and enlarged the BIOICE sampling grit, adding a genetic focus [3]. During the expedition IceAGE_RR (Icelandic marine Animals: Genetics and Ecology_Reykjanes Ridge hydrothermal vent activity, MSM75), a special focus was placed on the geology and ecology of the Reykjanes Ridge (RR) [3,4]. The RR is an extension of the Mid-Atlantic Ridge (MAR), separated by the Bight Fracture Zone, and elongates ~900 km

to the Icelandic peninsula [10]. In contrast to the rest of the MAR, the RR has an oblique orientation angle of 27°, and periodic adjustment by rift propagation formed its unique structure [10,11]. It is the longest V-shaped and volcanically influenced mid-oceanic ridge, and it's topography is mainly dominated by faults [10]. This unique area hosts various ecosystems such as hydrothermal vents [4,5,12], seamount chains built from pillow lava, and reefs build by cold-water corals [2,13,14].

Cold-water coral reefs form a diverse ecosystem in the deep sea [15–17]. During their growth, cold-water corals produce a three-dimensional hard substrate and provide a habitat for many other organisms such as ophiuroids, crustaceans, and polychaete worms [16–19]. Cold-water corals are found in all seas, but high densities have been reported in the northeast Atlantic; mostly in water temperatures ranging from 4 °C to 12 °C [13,15,17,18,20]. In 1996, Copley et al. [2] performed an extensive study on the fauna of the Reykjanes Ridge and reported species of the class Ophiuroidea that are continuously found with pieces of corals. Additionally, Buhl-Mortensen et al. [15] found associations of echinoderms, in particular ophiuroids and other organisms, with some species of gorgonians.

Ophiuroids are a widespread class and have been frequently reported from coral reefs [21–25]. They can occur in high biomass from the shore to the hadal trenches, and from tropical waters to the Arctic and Antarctic regions [26,27]. With 2123 described species [28], ophiuroids are the most speciose class within the echinoderms. They are common dwellers in the benthic environment and inhabit many habitats with contrasting characteristics [21,29–31]. Brittle stars often represent the largest proportion of the megabenthic community and are found in high abundance. Because of their widespread distribution, ophiuroids are a suitable class for genetic studies focusing on phylogeny or biogeography [21,30,32–38]. In benthic organisms, genetic connectivity is often linked to the reproductive strategy, which can differ among species. In ophiuroids, the most widespread reproduction type is spawning, wherein each individual can release up to 880,000 eggs into the water before developing via a planktotrophic development stage to a benthic juvenile, but it is also still unknown for many species [39–42]. Ophiuroid larvae can also be lecithotrophic, with the production of fewer eggs containing a yolk sack [39,40,43]. The third reproduction type is brooding. Here, about 2–2000 eggs are carried in the adult bursa slits and develop directly to the juvenile state without any pelagic phase [39,41,44].

In recent years, molecular tools and DNA barcoding in particular have provided a useful method for fast, efficient, and reliable species identification and discovery [45–48]. It is based on the concept that intraspecific diversity for the COI gene is lower than interspecific diversity. The resulting difference is called a "barcode gap" [48]. DNA barcoding not only shortcuts the difficulties of a morphology-based identification, e.g., when diagnostic characters are damaged during collection, but also connects the different stages of animal development [49]. A 658-bp region of the cytochrome c oxidase I (COI) gene has an effective marker as a species delimitation tool in different groups of marine organisms [37,46,50,51], particularly in brittle stars [33,34,38,52]. Currently, records are available for 10,798 ophiuroid specimens, representing 604 species on the Barcode of Life Data System, BoLD.

In this work, to create a baseline for further studies, we used an integrative approach based on morphological characters and DNA barcoding to study ophiuroid species associated with cold-water corals collected with a remotely operated vehicle (ROV) from Reykjanes Ridge south of Iceland. We further analyzed image data to embed the molecular study in an ecological view.

## 2. Materials and Methods

### 2.1. Morphological Identification

All specimens used in this study were sampled during the IceAGE_RR expedition MSM75 [3] in 2018. During the expedition, three cold-water coral reefs found at depths ranging from 239 m to 1579 m along the Reykjanes Ridge were investigated using the ROV Phoca (GEOMAR). The sampling areas used in this study are divided into the following

stations: Area 2 with stations 67-7, 188-4, 188-5, 188-6; Area 3 with stations 80-2, 80-5, 80-11, 111-6, 149-2, 149-3; and Area 4 with station 127-2 (Figure 1). All collected specimens were treated as described in Taylor et al. [4], fixated in 96% ethanol, and stored at −20 °C at the German Centre for Marine Biodiversity Research (DZMB), Senckenberg am Meer, Wilhelmshaven. Each specimen was then observed using a Leica M125 microscope and given an individual number (sample ID). A picture was taken from the oral and ventral sides (Leica EC3 Camera) and a tissue sample was collected from an arm segment for molecular analyses. The arm segment tissues were also fixated in 96% ethanol and stored at −20 °C. Morphological identifications were performed using Mortensen [53] and Paterson [54]. The morphology of the five species is represented in Figure 2.

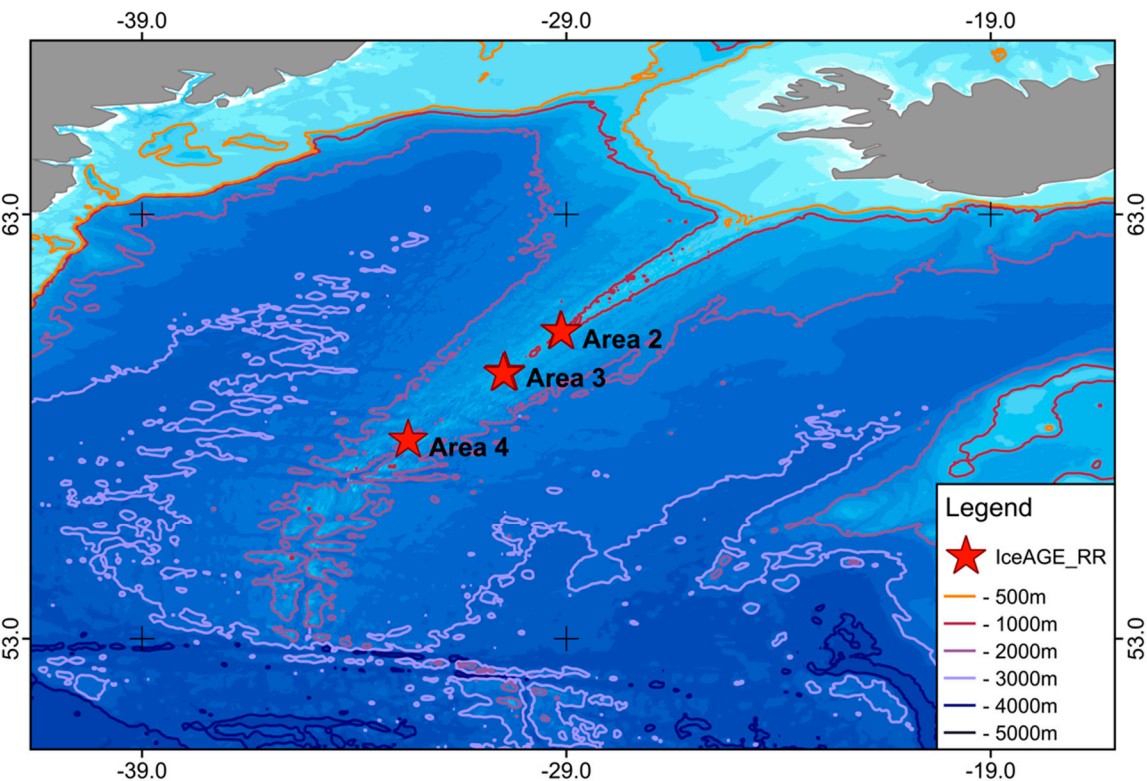

**Figure 1.** Sampling areas in the North Atlantic south of Iceland. Depth contours are displayed between 500 and 5000 m.

### 2.2. Molecular Species Delimitation

DNA extractions were carried out using 40 μL Chelex (InstaGene™Matrix) according to the protocol of Estoup et al. [55]. PCR protocols followed Christodoulou et al. [52].

The sequencing and sequence editing were conducted as they were in Khodami et al. [34]. Additionally, six sequences from the same morphospecies or genus were either downloaded from Genbank or completed by the DZMB databank and added to the dataset (KX459004.1; KU895176; KJ620586.1; KF663499.1; HQ946175.1; HQ919150.1; MT152642.1; DZMB54597; DZMB42400B; DZMB42400A; DZMB37435B; Supplementary Material Table S1). Sequences were aligned using MAFFT v7.017 [56] and blasted against GenBank. The alignment was exported as a FASTA file for further analysis. For each specimen, a picture and the sequence data were uploaded on the Barcode of Life Data System (BoLD, www.barcodinglife.org, accessed on 19 April 2022) in the project IARRO (Associated ophiuroid fauna on cold water corals of the Reykjanes Ridge; DOI: dx.doi.org/10.5883/DS-IAORR) and will also be available on GenBank (ON341454–ON341716).

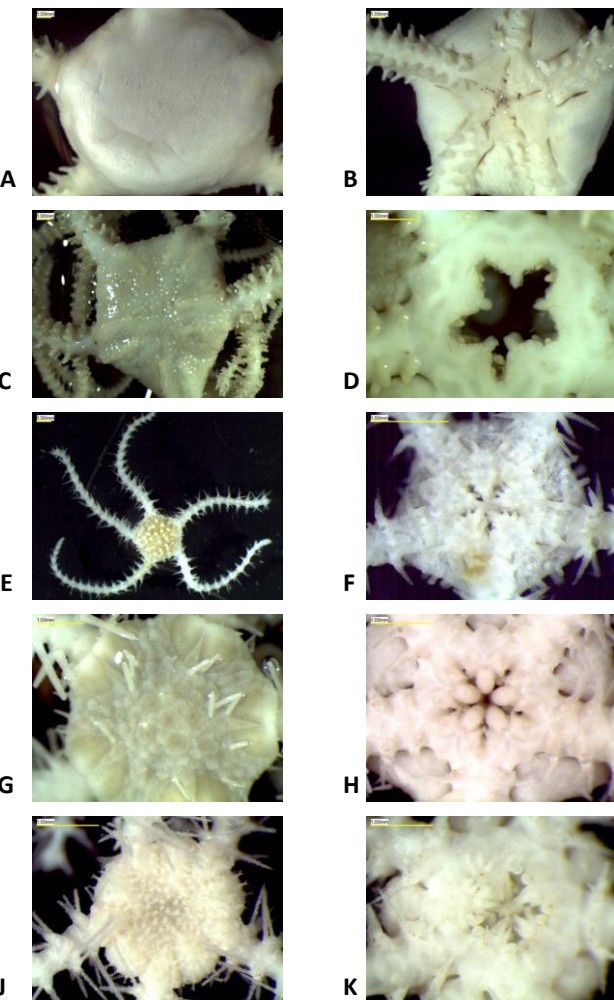

**Figure 2.** Morphology of the five species in dorsal and ventral views. (**A**,**B**): *Ophiomyxa serpentaria* Lyman, 1883; (**C**,**D**): *Ophiolebes bacata* Koehler, 1921; (**E**,**F**): *Ophiomitrella clavigera* (Ljungman, 1865); (**G**,**H**): *Ophiactis abyssicola* (M. Sars, 1861); (**I**,**J**): *Ophiacantha cuspidata* Lyman, 1879.

Three different methods for species delimitation were applied to assess the number of putative species. The first one, Barcode Index Number System [57], is a distance-based method. Within BoLD, the newly submitted sequences are compared with the sequences already available. They are clustered based on their molecular divergence using algorithms that aim to discover discontinuities between these clusters. A specific code (Barcode Index Number or BIN) is assigned to each cluster, either already existing or newly generated if submitted sequences do not match with sequences of already known BINs.

Automatic Barcode Gap Discovery (ABGD) is the second, distance-based method. It assigns the inserted specimens into species based on the pure distribution of pairwise differences (p-distance). The ABGD method initially calculates the indicative barcode gap to partition the sequences and gap detection, then continues recursively on previously obtained clusters to redefine partitions [48].

The third one is the Generalized Mixed Yule Coalescent (GMYC) method. It is a likelihood method for delimiting species by fitting within- and between-species branching models to reconstruct ultrametric gene trees. The model is described by Pons et al. and Monaghan et al. [58,59]. Species in this model are delimited by the descendent nodes of branches crossing the barcode gap threshold. This approach defines each species based on the most recent common ancestor on the phylogenetic tree and assumes that the most recent diversification event occurred before the oldest within-species coalescent event. The method should be implemented on a pre-analyzed Bayesian phylogenetic tree. Beast package

(Version 1.8.4, available on: http://beast.bio.ed.ac.uk, accessed on 20 February 2020) has been used to prepare the ultrametric rooted tree from COI sequences of ophiuroids, setting the tree prior to "Speciation: Yule Process" for $50^6$ generations with a sampling frequency of 5000 Generations. TreeAnnotator has been used to summarize the trees to a consensus tree with posterior probabilities. This tree has been used to run GMYC analysis by the SPLITS package (http://splits.r-forge.r-project.org, accessed on 20 February 2020) in the statistic program R (Version 3.3.2, available at: https://www.r-project.org, accessed on 5 January 2020). The genetically delimited species were examined for diagnostic morphological characteristics to confirm the species delimitation by morphological evidence in parallel to genetic analysis.

The distribution maps for each species are based on an extended data set, including the BIOICE data (Stöhr, previously unpublished material) and the public Ocean Biodiversity Information System (OBIS), and were completed by the stations of IceAGE_RR. The full distribution was added to QGIS (version 3.12, http://qgis.org, accessed on 30 November 2020) with which the distribution maps for each species were also created.

### 2.3. Image Data Analysis

Images used for ophiuroid assessment data generation were reviewed in random order to minimize time or sequence-related bias [60]. Specimens were identified to the collected morphospecies, measured using the BIIGLE 2.0 software [61], and assigned to a defined habitat. The data were then exported from BIIGLE and the statistical analyses were carried out with the following packages in R. The present-absence matrix was analyzed with the package "vegan"; a Nonmetric Multidimensional Scaling (nMDS) was carried out to determine if there were differences in the distribution of the ophiuroids among the defined habitats. The package "BiodiversityR" was applied to measure the Shannon index (H). The differences of the ophiuroid assemblages between the habitats were then calculated with a multilevel pairwise comparison using the package "pairwiseAdonis". Additionally, an indicator species was defined for each habitat with the package "indicspecies".

### 3. Results

### 3.1. Morphological and Molecular Species Delimitation

A total of 684 ophiuroids were picked by the ROV at three areas along the Reykjanes Ridge south of Iceland (Figure 1) and morphologically identified to the species level (Figure 2). DNA extractions were performed on 288 specimens. In total, 270 new sequences and 11 public sequences of 658 bp were included in the analysis based on COI. All, BIN, ABGD, and GMYC methods are congruent and delineate five species, which where morphologically identified as follows: *Ophiomitrella clavigera* (Ljungman, 1865); *Ophiomyxa serpentaria* (Lyman, 1883); *Ophiacantha cuspidata* (Lyman, 1879); *Ophiactis abyssicola* (M. Sars, 1861); and *Ophiolebes bacata* (Koehler, 1921). The morphology of the five species is represented in Figure 2. Figure 3 displays the maximum likelihood tree with the highlighted five species of this study. The maximum likelihood reconstruction has a high branch support with values greater than 92%. The mean p-distance within a species was 0.3% and the mean p-distance between species was 26.1%.

### 3.2. Ecological Analysis

The distribution map illustrated in Figure 4 presents different patterns for each species. It is striking that three species are only distributed south of Iceland (*O. serpentaria*, *O. cuspidata* and *O. bacata*). *Ophiomitrella clavigera* is mostly recorded from the south of Iceland but also from a single station in the shallow (between 100–300 m) north of Iceland. *Ophiactis abyssicola* has a wide distribution across the North Atlantic as well as in the Arctic region. The specimens of the IceAGE_RR project are, in most cases, the first record of these species from the RR.

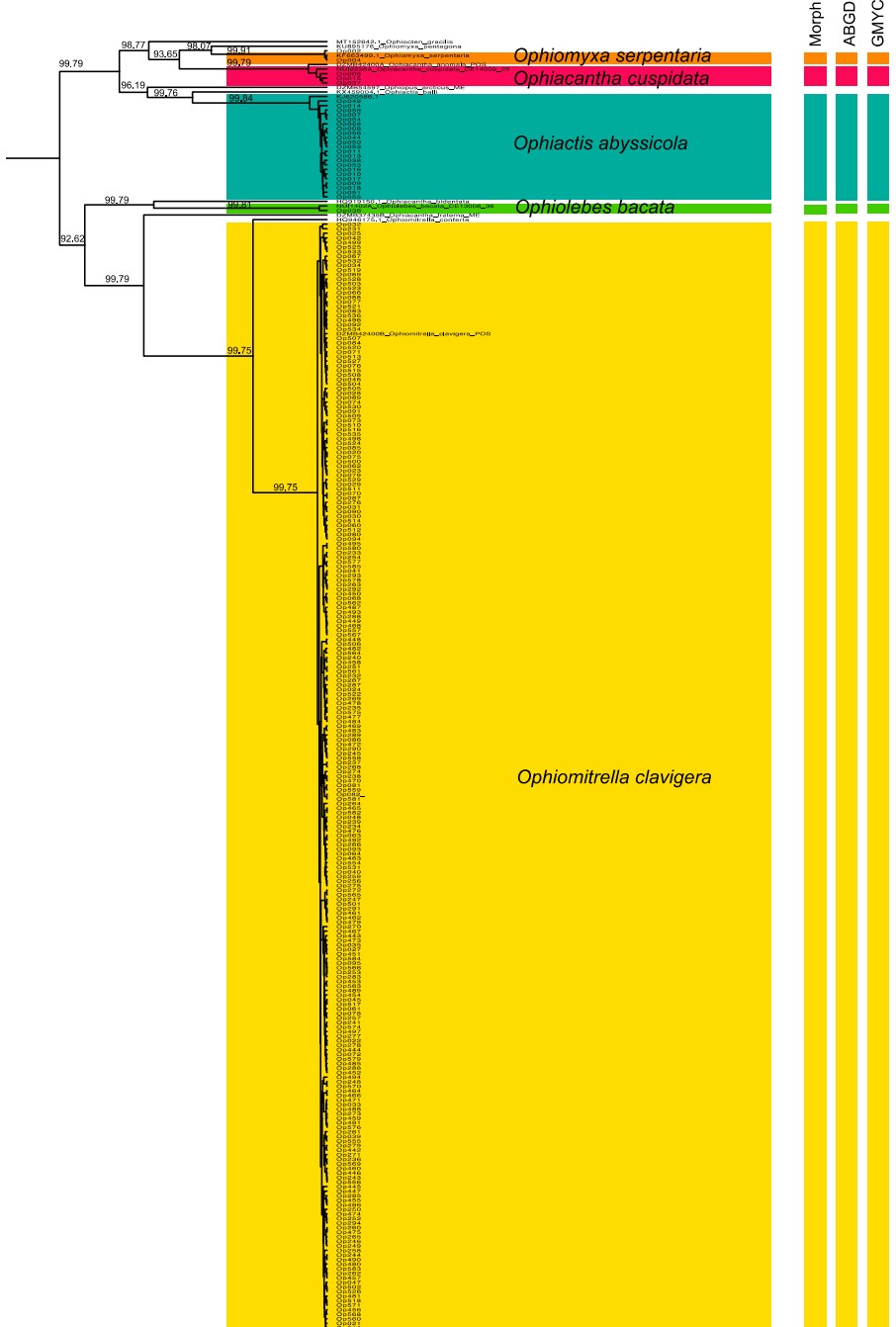

**Figure 3.** Gene tree of Bayesian analysis of COI (numbers on branches represent posterior probabilities >0.92) and species delimitation tools for COI fragment. The vertical bars represent alternative taxonomies, respectively supported by Automatic Barcoding Gap Discovery (ABGD) and General Mixed Yule Coalescent (GMYC), in addition to morphology. The colors are related to the identified clades.

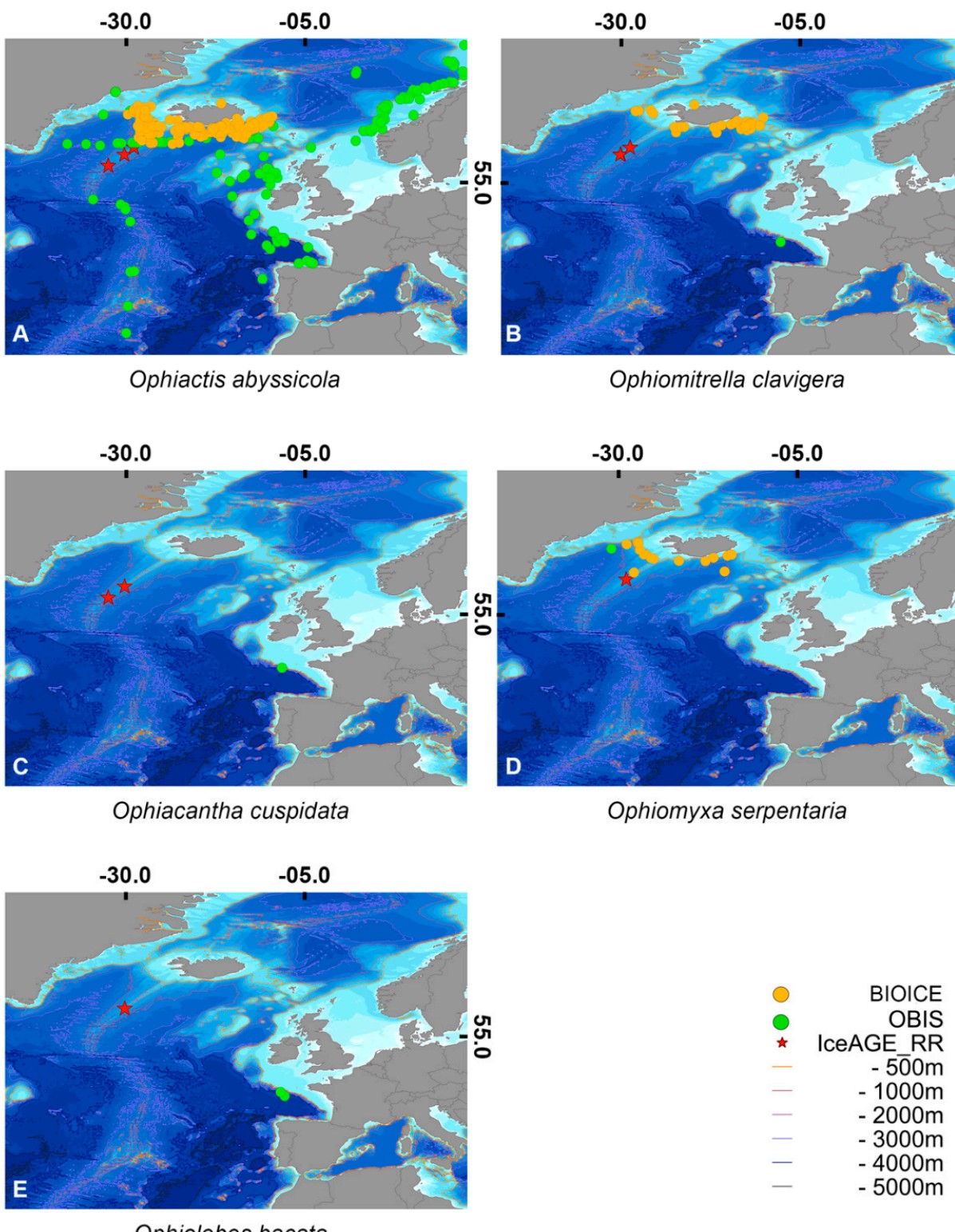

**Figure 4.** The maps represent the distribution of each ophiuroid species by reference (**A**) *Ophiactis abyssicola*, (**B**) *Ophiomitrella clavigera*, (**C**) *Ophiacantha cuspidata,* (**D**) *Ophiomyxa serpentaria*, (**E**) *Ophiolebes bacata*. The red stars indicate the stations where the ophiuroids were sampled during IceAGE_RR. The green dots illustrate the stations that were reported by OBIS (accessed 24 September 2021). The yellow dots are records from BIOICE material.

A total of 5174 ophiuroid individuals were observed in the four ROV dives during the present study. The ophiuroids were classified into five morphospecies: *O. abyssicola*, *O. clavigera*, *O. cuspidata*, *O. serpentaria*, *O. bacata*, and "other" for the specimens that couldn't be categorized into a morphospecies. Only *O. abyssicola*, *O. clavigera*, and *O. cuspidata* were found in the analyzed images. The habitats are pictured in Figure 5 and were defined as: "Pillow", "Coral", "Pillow + Coral", "Pillow + Coral rubble", and "Pillow + Sediment." Distribution of the ophiuroid species differed with each habitat (Figure 6A–E). *Ophiomitrella clavigera* was mostly found on corals but rarely in the other habitats. The other species were equally distributed among the remaining habitats. The comparison of the species assemblages is displayed in an nMDS plot of the ophiuroid diversity divided into the different habitat types (Figure 6F). Generally, a separation can be seen between the habitat type "coral" and the rest of the habitat types. The Shannon index (H) between all habitats is 0.26 ($p$ = 0.32). However, the pairwise adonis indicates a significant ($p$ = 0.01 *) difference among all habitats, except for the comparison between the habitats "Pillow" and "Pillow + Sediment" ($p$ = 0.52). The test for the indicator species per habitat calculated *O. clavigera* (0.669; $p$ = 0.005 **) for the habitat "Coral", *O. cuspidata* (0.308; $p$ = 0.03 *) for the habitat "Pillow + Coral Rubble" and *O. abyssicola* (0.765; $p$ = 0.005 **) for all the other habitat types.

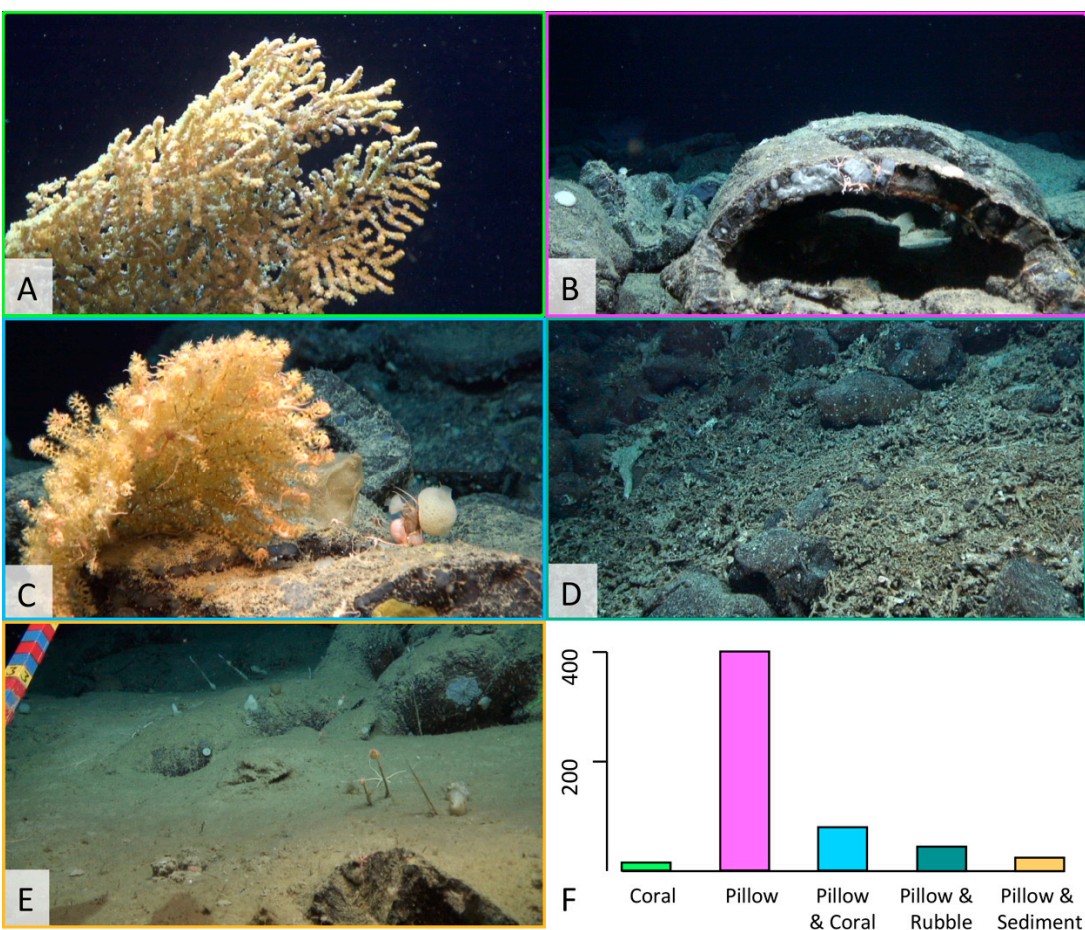

**Figure 5.** (**A**–**E**): The images demonstrate a representative of each defined habitat type. (**A**): Coral; (**B**): Pillow; (**C**): Pillow + Coral; (**D**): Pillow + Coral Rubble; (**E**): Pillow + Sediment. (**F**): The bar plot displaying the proportional occurrence of each habitat.

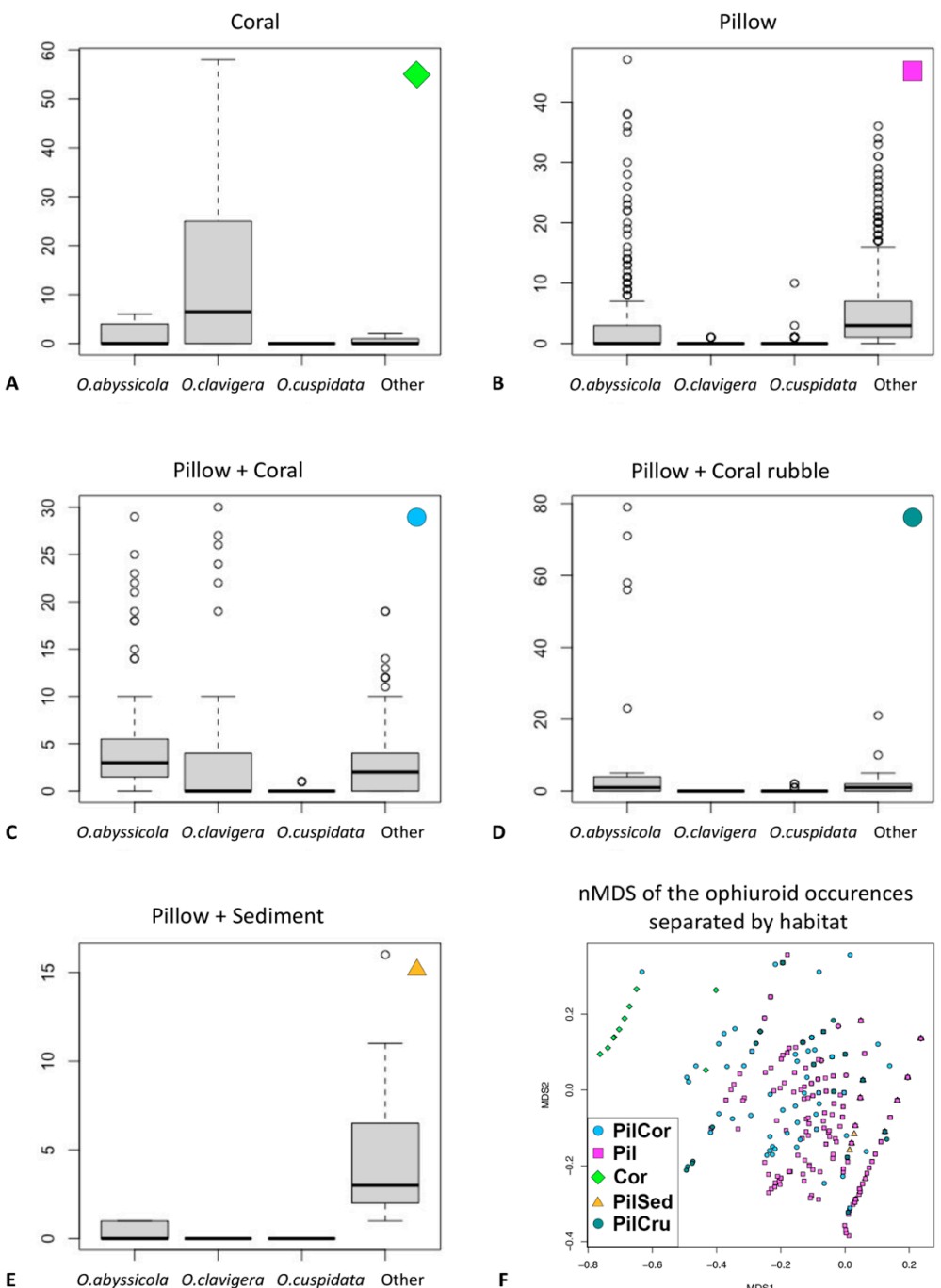

**Figure 6.** (**A–E**): The boxplots illustrate the proportional abundance of *Ophiomyxa serpentaria* Lyman, 1883; *Ophiolebes bacata* Koehler, 1921; *Ophiomitrella clavigera* (Ljungman, 1865); *Ophiactis abyssicola* (M. Sars, 1861); and *Ophiacantha cuspidata* Lyman, 1879 relatied to each habitat: (**A**): "Coral" (Cor); (**B**): Pillow" (Pil); (**C**): "Pillow + Coral" (PilCor), (**D**): "Pillow + Coral Rubble" (PilCru); (**E**): "Pillow + Sediment" (PilSed). (**F**): The distribution of the ophiuroid fauna depending on the habitat presented by Nonmetric Multidimensional Scaling (nMDS).

## 4. Discussion

### 4.1. Ophiuroids Distribution According to Seafloor Heterogeneity

The ophiuroid species in this work were all found on cold-water corals and recorded in previous studies from the North Atlantic [62–64]. *Ophiomyxa serpentaria* and *O. abyssicola* were found during different expeditions in the northeast Atlantic [63,64] and particularly on the RR [2]. There is a significant ($p$ = 0.01) separation of the faunal composition in

the nMDS plot between the different habitats, which may be explained by differences in food supply, surfaces, and structures of the habitats [65–70]. Particularly interesting is the indicator species from the coral habitat, which was identified as *O. clavigera* (0.669; *p* = 0.005). This species was observed to exclusively use gorgonians as hosts. In contrast to other coral-associated ophiuroids that sit alone or in small groups on the corals (e.g., reported for *O. oedipus* Lyman, 1879 or some species of the genus *Asteroschema* Örsted and Lütken, 1856 [22,23,71]), the specimens of *O. clavigera* aggregate with hundreds of individuals per coral. Additionally, their brooding lifestyle leads to the suggestion that there is a higher gene flow within these aggregations than between them. As with *O. clavigera*, *O. abyssicola* has been reported in association with different hosts, like corals or sponges, but it also occurs in high densities on the seafloor [2,19,63,64]. To use corals as hosts serves different advantages. One advantage is the species can expand their habitat from the seafloor to a higher level and reach new food resources [15,19,22] as the branches of the corals are often in a strong current with different food sources, like planktonic organisms as copepods or dead particles from the sea surface. Whereas many ophiuroids are known to be suspension feeders with an unspecified diet, they are observed with some arms entwined around the coral branch and the other arms within the current. Furthermore, the higher level can enhance larval dispersal when the eggs are released into the water currents [2,15,19,25,72]. Conversely, ophiuroids can use corals as a shelter from strong currents or predators [15,19]. However, dead coral, classified as coral rubble, was inhabited by various ophiuroid species, especially by *O. cuspidata*, which was the indicator species for this habitat (0.308; *p* = 0.03). The remnants of the corals still form a diverse habitat with hard substrates, holes, and hollows that provide shelter for the ophiuroids and is further inhabited by smaller invertebrates that serve as food supply [18,73]. The most abundant habitat was the lava pillows that form a smooth hard substrate surface with some cracks and folds in between. Mostly, the indicator species *O. abyssicola* was living inside these formations extending their long and spiny arms in the water column, which suggests a suspension feeding lifestyle. Additionally, individuals were also spotted a few times sitting on corals with two to three arms attached to the branch and the others extended in the water column. However, an association with a special host coral couldn't be identified, as they were seen on varying coral species as well as highly abundant on the seafloor.

*4.2. Deep Icelandic Ophiuroid Fauna*

The deep-sea ophiuroid fauna around Iceland has been studied for decades taxonomically [42,54,62,74,75]. In 1985, Paterson published a book about "The deep-sea Ophiuroidea from the North Atlantic Ocean" containing approximately 120 species living below 1000 m [54]. In recent years, new taxonomic discoveries have been made [26,76], and more information about genetic diversity has been established [22,30,33,34,77,78]. The study by Copley et al. [2] focused on the faunal community living on the RR. It suggested a different bathymetric range for species with a split between 800 m and 1000 m, because of the transition of the two water masses occurring at this depth (SMW and UNADW). *Ophiactis abyssicola*, *O. clavigera,* and *O. serpentaria* were recorded along the RR across the whole depth range, so differentiations of the bathymetric depth cannot be supported in this study. The contrast between the results presented by Copley et al. [2] and our work is not surprising. Deep-sea environments are among the least studied on our planet [16,79] and bias in the sampling as well as cryptic speciation could be possible [52,80–82]. A comparable approach of barcoding the Icelandic ophiuroid fauna was caried out from a more northern area by Khodami et al. [34]. These authors found a completely different ophiuroid community than described in this study. Due to different abiotic factors (e.g., soft sediment instead of hard substrate, differences in depth and temperature) as well as the different sample set up (ROV vs. AGT, EBS, and TAD) the differences between the studies reflect the diversity of the deep-sea ophiuroid fauna around Iceland and that a wide sample set up with different sampling methods should be used.

*4.3. Species Delimitation and DNA Barcoding*

Over the last few decades, DNA barcoding has been increasingly used for species delimitations and has proven to be an effective method for many organisms, including ophiuroids [34,37,38,83]. The species from the present data set could also be clearly distinguished from each other by the established DNA barcoding method, so we focused more on the way of life than on the genetic analysis. However, five BINs could be assigned to our studied material. Three of the BINs are identified to the species level and publicly accessible, whereas the other two are unique, held in private datasets, or lack detailed identification. This indicates that genetic information of deep-sea megabenthic fauna is still quite unknown [52,84–87], and this dataset acts as a supplement to this knowledge. Working at such a small regional scale certainly does not mirror the complete variation within each species, and indicates the limits of DNA barcoding. It only provides partial information due to being limited to just one region in the mitochondrial gene [88]. However, this method has proved useful to extend background knowledge of taxonomy, population genetics, and molecular phylogenetics [89]. DNA barcoding reduces the difficulties in identifying damaged specimens and can overcome identification problems with morphologically highly variable species, and is helpful in connecting the larvae to the adults [49]. In this work, we had different developmental stages of *O. clavigera*, and DNA barcoding supported the result that all of them belong to this species. The mean intra- and interspecific p-distances (0.3% and 26.1%) were in the known range for echinoderms and do not present any signs of cryptic speciation [34,38]. It is important to point out that the taxonomic and molecular diversity in this study does not reflect the complete diversity of ophiuroid species from Icelandic and adjacent waters, but reflects only a small representation. For example, the study published by Khodami et al. [34] reported a completely different ophiuroid community with a higher molecular and taxonomic diversity. The authors further reported that results in morphology can differ from genetic evidence and that closer related species were hard to delimit, as presented for the genus *Ophiacantha*. Cryptic speciation or recently separated species are still problematic in genetic analyses [30,33,34,82]. However, the five species from this study, belonging to different genera, did not present any cryptic speciation and could be separated well from each other. It is worth highlighting that the genetic methods were congruent with the morphological identification.

**5. Conclusions**

Five ophiuroid species were successfully and consistently identified in all methods, using morphological characters combined with the COI marker. This study highlights the exclusive association of the brittle star species *Ophiomitrella clavigera* with non-species-specific gorgonian cold-water corals. The combination of video data with morphological and molecular information from actual sampled specimens highlights the possibility to better understand the variety in ecology and geography that is often mirrored in the genetic diversity. We need to understand the species in connection with their environment to be able to draw conclusions from our morphological and genetic results. Video surveys help to observe specimens in their actual environment and supplement the knowledge from classical sampling methods.

**Supplementary Materials:** The following supporting information can be downloaded at: https://www.mdpi.com/article/10.3390/d14050358/s1, Table S1: Table of additional sequences used in the dataset.

**Author Contributions:** A.E., S.B. and P.M.A. designed the study, supervised all steps of the project, analyzed the data, and drafted the manuscript. S.B. organized the sampling and provided the biological material. A.E. performed the COI sequencing. A.E. and P.M.A. analyzed the COI data. S.S. did the morphological species identification and provided the BIOICE distribution data of ophiuroid species. A.E. and J.T. analyzed the image data. All authors have read and agreed to the published version of the manuscript.

**Funding:** The sampling of the specimens was funded via the German Science Foundation (DFG grant MERMET17-15).

**Institutional Review Board Statement:** Not applicable.

**Data Availability Statement:** The sequence and meta data of the specimens were deposit in GenBank and BoLD (DOI).

**Acknowledgments:** We would like to thank the following people for their key contributions to our manuscript: the crew of MS Merian during IceAGE_RR (MSM75) cruise; Tim O'Hara (Victoria Museum Melbourne), who helped to check the sequencing data and provided some complement sequences for not collected species; Antje Fischer and Karen Jeskulke (TAs DZMB HH) for their support in sorting the samples; and Sahar Khodami (DZMB Wilhelmshaven) for the help in the lab.

**Conflicts of Interest:** The authors declare no conflict of interest. The research was conducted in the absence of any commercial or financial relationships that could be construed as a potential conflict of interest.

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
