# Peer review of "DNA Barcoding of Cold-Water Coral-Associated Ophiuroid Fauna from the North Atlantic"

_diversity, doi:10.3390/d14050358_

Round 1

Reviewer 1 Report

Dear, Author.

This article is quite impressive in the field of taxonomic studies for echinoderm.
All approaches to the analysis (e.g., sampling with ROV, classical morphological taxonomy, and DNA barcoding) suggest a next-gen taxonomic co-working.

But, some minor spell-checking needs before publishing (e.g., comma in numbers). Please, check specific comments in a PDF file.

It is my honor that reviewed this article at this time and respect all your works in this article.

Thank you.

Author Response

Dear Reviewer,

Thank you for your comments and time. We adapted all the changes you made. To answer your question about the way to show the posterior possibilities, I was following the paper of Khodami, 2014 which had a similar study in this field. They also showed the posterior possibilities in percentages.

Thank you for your kind words.

Kind regards

Reviewer 2 Report

As noted, faunas in the deep sea are the most poorly studied habitats on Earth, in fact, we probably know more about the surface of Mars than the benthic habitats of the deep sea on Earth. Thus, in general, this is a very important contribution to better understand deep sea habitats.

In particular, this is a well-written manuscript using a very interesting, relatively recent and significant method (DNA barcoding coupled with morphological study) to study marine biodiversity. The manuscript is straightforward. As Earth enters into its next great period of extinction, it is vital to document all habitats in order to document biodiversity losses as well as to understand the potential origin of new faunas that radiate during recovery. The deep sea is a potential source for replacement taxa for extinctions that occur in shallower-water settings. Documenting this fauna is quite important

The methods are sound, and the conclusions are well explained. I have a few comments on the text for consideration by the authors (see attached pdf).

Again, this is an excellent paper, and I urge publication as soon as possible.

As noted above, a pdf with a few editorial remarks is attached for consideration by the authors.

This is a self-serving comment (and I apologize); but on p. 10 where the authors discuss ophiuroids attached to gorgonians and various hard corals, they could mention that this is an example "secondary tiering" (see Bottjer and Ausich 1987. Phanerozoic development of tiering in soft substrata suspension-feeding communities. Paleobiology 12:400-420.) 

Author Response

Dear Reviewer,

Thank you for your review. We consider all the changes you suggest in the PDF. Thank you for your detailed message and the appreciation of this study. We agree that this kind of studies are important to understand the deep-sea and its habitats better and can thus contribute to the protection of the ecosystem. Unfortunately, we could not include the reference in our manuscript, as we cannot confirm the statement as it stands.

Kind regards

Reviewer 3 Report

The presented MS is interesting and well illustrated. However, I recommend some minor corrections and additions (see also the PDF).

INTRODUCTION

It would be very nice to provide a map illustrating what is described in the lines 32-50.

MATERIALS AND METHODS

Line 123: GenBank numbers should be inserted. The table with these numbers should be combined with Table A1 of Supplementary Material and this general table should be placed right to the Material and Methods paragraph.

In the Table A1, I have not found the number MT152642.1 Ophiocten gracilis, and the number NU12256A Ophiacantha cuspidata. They are present in Figure 3, however they are missing in the table. The number KJ620586.1 is not accompanied with the species name in the Figure 3. The number DZMB30291A   Ophiacantha aculeata is present in the table, but it is missing in the Figure 3.

How many individuals of each species were used for the genetic and ecological analysis? I have not found this data.

I did not find clear information about the depths at which the animals were sampled. The only information is the map in Fig. 1. However, this is very unclear information. It is necessary to give a table with station numbers, collection depths and the quantity of collected specimens of the studied ophiuroids.

RESULTS

All five studied species belong to five different genera. Therefore, it is unacceptable to shorten generic names to one letter in the text. Such a reduction is possible only if we are consistently talking about two different species of the same genus. In all other cases, generic names should be indicated in full along with specific ones.

The first three sentences of the Results (lines 178-181) relate more to the Materials and Methods than to the Results. The Results paragraph may begin from "All, BIN, ABGD and GMYC methods are congruent and delineate five species...".

Figures 1 and 4: The depth contours are poorly visible (except 500 m and 1000 m). If possible, it is better to mark them with bright colors contrasting with the background.

Figure 2: The names of species in the figure caption should be in italics.

Figures 4 and 6: It is better to give the full names of species (including the names of genera) under the maps and boxplots.

Figure 5: Minor corrections in the figure caption (see PDF).

It is unclear why abbreviations Cor, Pil, PilCor, PilCru, PilSed are needed for different habitat types. They were used only once in the Figure 6F. Throughout the text and in the captions to the figures, the full names of the habitat types are used (“Coral”, “Pillow”, “Pillow + Coral”, “Pillow + Coral rubble”, “Pillow + Sediment”). It seems to me that these abbreviations are not needed at all; they only overload the perception of the proposed information.

Lines 217, 218: The sentence "A total of 5,174 ophiuroid individuals were observed in the four ROV dives during the present study" is for Materials and Methods.

Lines 221-223: The definition of the habitat types here repeats the lines 163-164 of Materials and Methods. It is needed to leave this definition only in one place (preferably in the Results).

DISCUSSION

The discussion is interesting, but more like a literature review. It is necessary to emphasize the conclusions drawn on the basis of the authors' own data set out in the Results.

It is needed a clear analysis of the distribution of different species on different substrates.

To what extent does the morphological definition coincide or does not exactly coincide with the genetic method? Were there any errors in morphological determination corrected by genetic methods? According to Fig. 3, the morphological definition in all cases coincided with the genetic one. Is that so? It is necessary to highlight this thesis in the Results and in the Discussion.

REFERENCES

The names of species should be in italics, aren't they?

Please either capitalize all the major words in the English titles or start those words (excepting the first) with lower case letters (preferably). Please do not mix these two styles; select the one and use it consistently.

Author Response

Dear Reviewer,

Thanks a lot for your review. We appreciate your detailed comments and suggestions. Most of your proposed points were considered. You can see our answers in the attached file. We hope we were able to clear all ambiguities.

Kind regards
